

# Genome-wide identification, evolution, and expression of the *SNARE* gene family in wheat resistance to powdery mildew

Guanghao Wang[1,2], Deyu Long[3], Fagang Yu[2], Hong Zhang[1,2], Chunhuan Chen[1,2], Yajuan Wang[1,2] and Wanquan Ji[1,2]

[1] Shaanxi Research Station of Crop Gene Resources and Germplasm Enhancement, Ministry of Agriculture, Yangling, Shaanxi, China
[2] College of Agronomy, Northwest A&F University, Yangling, Shaanxi, China
[3] College of Life Sciences, Northwest A&F University, Yangling, Shaanxi, China

## ABSTRACT

SNARE proteins mediate eukaryotic cell membrane/transport vesicle fusion and act in plant resistance to fungi. Herein, 173 SNARE proteins were identified in wheat and divided into 5 subfamilies and 21 classes. The number of the *SYP1* class type was largest in *TaSNAREs*. Phylogenetic tree analysis revealed that most of the SNAREs were distributed in 21 classes. Analysis of the genetic structure revealed large differences among the 21 classes, and the structures in the same group were similar, except across individual genes. Excluding the first homoeologous group, the number in the other homoeologous groups was similar. The 2,000 bp promoter region of the *TaSNARE* genes were analyzed, and many W-box, MYB and disease-related cis-acting elements were identified. The qRT-PCR-based analysis of the *SNARE* genes revealed similar expression patterns of the same subfamily in one wheat variety. The expression patterns of the same gene in resistant/sensitive varieties largely differed at 6 h after infection, suggesting that SNARE proteins play an important role in early pathogen infection. Here, the identification and expression analysis of SNARE proteins provide a theoretical basis for studies of SNARE protein function and wheat resistance to powdery mildew.

Corresponding authors
Yajuan Wang, wangyj7604@163.com
Wanquan Ji,
jiwanquan2008@126.com

## INTRODUCTION

SNARE (soluble N-ethylmaleimide sensitive factor attachment protein receptor) proteins are employed for a significant number of vital transport processes, as they mediate the fusion of the membranes of cargo-containing small shuttles, which are referred to as vesicles, and target membranes (*Lipka, Kwon & Panstruga, 2007*). These proteins are involved in vesicle membrane fusion and are responsible for transport in the endomembrane system, as well as for endocytosis and exocytosis. According to their functions, SNARE proteins can be divided into vesicle-associated (v-SNARE) and target-membrane-associated (t-SNARE) proteins (*Söllner et al., 1993*). Alternatively, SNAREs can be grouped as Q-SNAREs and R-SNAREs. These proteins have either conserved glutamine or conserved arginine residues in the center of the SNARE domain, and Q-SNAREs can be further subdivided into

Qa-SNAREs, Qb-SNAREs, and Qc-SNAREs (*Bock et al., 2001*). SNAP-25-like proteins comprise Qb-SNARE and Qc-SNARE motifs (*Schilde et al., 2008*). R-SNAREs have an either short or long N-terminal regulatory region, further subdividing them into brevins and longins (*Lipka, Kwon & Panstruga, 2007*). Previous studies have shown the existence of 60 SNARE proteins in *Arabidopsis thaliana*, 57 *SNARE*s in *Oryza sativa*, 69 *SNARE*s in *Populus trichocarpa* (*Lipka, Kwon & Panstruga, 2007*), 27 *SNARE*s in wheat (*Gaggar, Kumar & Mukhopadhyay, 2020*) and 21 *syntaxins* in *Solanum lycopersicum* (*Bracuto et al., 2017*). In addition, Sanderfoot revealed the evolution of eukaryotic SNAREs (*Sanderfoot, 2007*).

The plant cell endomembrane secretion pathway plays an important role in the interaction between plant cells and microbes (*Snyder & Nicholson, 1990*; *Walther-Larsen et al., 1993*). Plant cells are capable of identifying pathogen-associated molecular patterns through surface receptors, and cell surface receptor proteins recognize signal peptides. It was shown that the processing and positioning of these receptors occur through the protein secretion pathway (*Wang & Dong, 2011*). The autoimmunity of plants to infiltration by powdery mildew fungi is accomplished by targeting the cell wall with papillary factors, including purines, cytoplasmic components, extracellular membrane components and SYP121/PEN1 (*Nielsen et al., 2012*). In *Arabidopsis*, *PEN1* (*SYP121*) and its closest homolog, *SYP122*, appear to have a fundamental function in secretion and specific defense-related functions at the plant cell wall (*Assaad et al., 2004*; *Collins et al., 2003*). Similarly, HvROR2 (*Collins et al., 2003*) and SiPEN1 (*Bracuto et al., 2017*) were shown to be associated with defense against powdery mildew fungi. The AtSYP121/AtPEN1-AtSNAP33-AtVAMP-721/722 protein complex can assist cell emesis at sites of fungal invasion (*Douchkov et al., 2005*; *Kwon et al., 2008*; *Lipka, Fuchs & Lipka, 2008*; *Wick et al., 2003*). In addition, AtSEC11 modulates PEN1-dependent vesicle trafficking by dynamically competing for PEN1 binding with VAMP721 and SNAP33 (*Karnik et al., 2013*).

MdSYP121 affects the pathogen infection process in apples by regulating the salicylic acid (SA) pathway and the oxidation–reduction process (*He et al., 2018*). The SYP4 group regulates both secretory and vacuolar transport pathways and the related extra cellular resistance to fungal pathogens (*Uemura et al., 2012*). NbSYP132 may act as a cognate target SNARE protein receptor and positively regulate the exocytosis of vesicles containing antibacterial pathogenesis-related (PR) proteins (*Kalde et al., 2007*). Silencing of *StSYR1* enhances the resistance of potato to *Phytophthora infestans* (*Eschen-Lippold et al., 2012*).

OsVAMP714 can regulate disease resistance to blast in rice, but OsVAMP7111 cannot. Furthermore, OsVAMP714 overexpression promotes leaf sheath elongation (*Sugano et al., 2016*). Ectopic expression of *AtBET12* had no inhibitory effect on general ER-Golgi anterograde transport but led to intracellular accumulation of PR1 (*Chung et al., 2018*). GOS12 was an essential host factor for plasmodesmata (PD) targeting of the P3N-PIPO protein to defend against soybean mosaic virus (*Song et al., 2016*). *AtMEMB12* was targeted by miR393b* to modulate the exocytosis of antimicrobial PR1 (*Zhang et al., 2011*). AtSyp71 was a host factor that was essential for successful viral infection, mediating the fusion of virus-induced vesicles with chloroplasts during *TuMV* infection (*Karnik et al., 2013*). *OsSEC3A* enhances rice resistance to *Magnaporthe oryzae* by negatively regulating the

pathogenesis and expression of SA synthesis-related genes (*Ma et al., 2018*). *TaNPSN11*, *TaNPSN13*, and *TaSYP132* have diverse functions in the prevention of *Pst* infection and hyphal elongation (*Wang et al., 2014*).

Wheat is an important food crop in the world, but its output is subject to various severe biological and abiotic stresses. Wheat powdery mildew may occur in each growth period of wheat, which is mainly manifested in the leaves and cases. The main symptom is the appearance of a white powdery mildew layer on the leaf surface, which gradually expands and unites to constitute an oval mold and an irregular mold layer, as well as a layer of powdery substance (conidia) on the surface. In severe cases, a gray mold layer will be established on the leaves and black particles will appear. After the stems and leaves are infected, the wheat was prone to lodging and shrinking without heading. Discovering and using resistance genes is an environmentally friendly and economical way to resist wheat powdery mildew. Herein, 173 SNARE proteins were identified in wheat. Phylogenetic tree analysis revealed that most of the SNAREs were distributed in 21 classes. *TaSNARE* genes include many W-box, MYB and disease-related cis-elements in the promotor region. The expression patterns largely differed at 6 h after infection with powdery mildew. This study aimed to develop a better understanding of the identification, evolution, and expression of *SNARE*s and explore the relationship between wheat *SNARE*s and powdery mildew.

## MATERIALS AND METHODS

### Identification of *TaSNARE* genes

The wheat genomes and annotations used were from the newest IWGSC (International Wheat Genome Sequencing Consortium) v1.0 (https://wheat-urgi.versailles.inra.fr/Seq-Repository). The hidden Markov models (HMMs) of the SNARE (PF05739), Syntaxin (PF00804), longin (PF13774), Synaptobrevin (PF00957), SEC20 (PF03908), V-SNARE-C (PF12352), V-SNARE (PF05008) and USE1 (PF09753) motifs were downloaded from the Pfam database (http://pfam.sanger.ac.uk/). The wheat SNARE protein sequences were analyzed with HMMER 3.0 (http://hmmer.janelia.org/) as the query and the default parameters ($E < 0.01$). All presumptive *SNARE* genes were retained and confirmed using the Pfam database and the NCBI conserved domain database (https://www.ncbi.nlm.nih.gov/Structure/cdd/wrpsb.cgi). The molecular weights (MW) and protein isoelectric point (pI) of the *TaSNARE* genes were obtained using the tools from the ExPASy website (https://www.expasy.org/).

Multiple alignments of SNARE proteins were performed using ClustalW (*Larkin et al., 2007*) in MEGA 7.0 (http://www.megasoftware.net/). Phylogenetic analyses were performed using the NJ (neighbor-joining) method in MEGA 7.0 (*Kumar, Stecher & Tamura, 2016*) with 1,000 bootstrap resampling, the Jones-Taylor-Thornton (JJT) model (*Jones, Taylor & Thornton, 1992*), and the pairwise deletion option. Gene Ontology (GO) enrichment analysis of SNAREs was implemented using the clusterProfier R package (R version=4.0.3).

### Exon/intron structure analysis, conserved motif identification and cis-acting elements analysis

The gene structure provides important information, including disaggregated and evolutionary relationships among gene families. The *SNARE* genomic sequences and CDS sequences extracted from the plant database were compared in the gene structure display server program to determine the exon/intron organization of *SNARE* genes. The default parameters were used in the Multiple Em for Motif Elicitation (MEME) (http://meme-suite.org/) program for the identification of conserved protein motifs and a maximum number of 15 motifs. Promoter of *the SNARE* gene was used to analyze the cis-acting elements as previously described by *Sun et al. (2017)*.

### Chromosomal locations and gene collinearity analysis

The physical chromosome locations of all SNAREs were obtained from the gff3 files of wheat databases. TBtools (https://github.com/CJ-Chen/TBtools) software was adopted to visually map the chromosomal location. Gene duplication events were analyzed using the Multiple Collinearity Scan toolkit (MCScanX: http://chibba.pgml.uga.edu/mcscan2/). To exhibit segmentally duplicated pairs and orthologous pairs of *SNARE* genes, we used Dual Systeny Plotter software (https://github.com/CJ-Chen/TBtools) to drawn collinearity maps.

### Fungus and wheat materials

The wheat–*Ae. geniculata* disomic addition line NA0973-5-4-1-2-9-1 (CS-SY159 DA 7M$^g$, (CS)/*Ae. geniculata* SY159//CS)) was used (*Wang et al., 2016*). 'Shaanyou 225' was the powdery mildew susceptible control variety. The wheat–*Ae. geniculata* disomic addition line TA7661 (CS-AEGEN DA 7M$^g$) was kindly provided by Dr. Friebe BR and Dr. Jon Raupp of the Department of Plant Pathology, Throckmorton Plant Sciences Center, Kansas State University, Manhattan, KS, USA. The powdery mildew isolates E09 was maintained on the susceptible wheat 'Shaanyou 225'. All plants were cultured in a growth chamber with soil at 18 °C under a 16 h light/8 h dark photoperiod. The 14-day-old seedlings were inoculated with powdery mildew conidia from 'Shaanyou 225' seedlings infected 10 days previously using the dusting method. Wheat leaves were collected at 0 h, 6 h, 12 h, 24 h, and 48 h after powdery mildew infection, and quickly put into cryopreservation tubes and stored in liquid nitrogen. The leaves were used for the next step of RNA extraction and q-PCR experiments. The method by *Wang et al., (2020)* was used for CS (Chinese Spring), 7M CH (NA0973-5-4-1-2-9-1), 7M US (TA7661) and 'Shaanyou 225' to identify powdery mildew.

### RNA-seq expression analysis of *SNARE* genes

To further understand the function of the *SNARE* gene, we investigated the reported RNA-seq data, including the developmental timecourse in five tissues (*Choulet et al., 2014*), grain layers (*Pearce et al., 2015*), grain layer developmental timecourse (*Pfeifer et al., 2014*), senescing leaves timecourse (*Pearce et al., 2014*), photomorphogenesis of DV92 and G3116 (*Fox et al., 2014*), and drought and heat effects (*Liu et al., 2015*). The data were analyzed using MeV (Multi Experiment Viewer) software. Data obtained from the RNA-seq

expression atlas were normalized based on the mean expression value of each gene in all tissues/organs analyzed and clustered by the hierarchical clustering method.

The developmental time course in five tissues includes all of the wheat stage, as follows (*Zadoks, Chang & Konzak, 1974*): seeding (first leaf through coleoptile, Zadoks Scale 10, Z10), three leaves (three leaves unfolded, Z13), three tillers (Main shoot and 3 tillers, Z23), spike at one cm (pseudostem erection, Z30), two nodes (2nd detectable node, Z32), meiosis (flag leaf ligule and collar visible, Z39), anthesis (1/2 of flowering complete, Z65), 2 days after anthesis (DAA) (Kernel (caryopsis) watery ripe, Z71), 14 DAA(medium Milk, Z75), and 30 DAA (soft dough, Z85). The grain layers contained three parts at 12 days post-anthesis (DPA): the outer pericarp, inner pericarp, and endosperm. The grain layer developmental timecourse included the following seven-stages: 10 DPA whole endosperm, 20 DPA whole endosperm, 20 DPA starchy endosperm, 20 DPA transfer cells, 20 DPA aleurone, 30 DPA starchy endosperm, and 30 DPA aleurone plus endosperm. The senescing leaves timecourse contains three stages: heading date (HD), 12 DAA and 22 DAA. The photomorphogenesis of the wild winter wheat *T. monococcum ssp. aegilopoides* (accession G3116) and the domesticated spring wheat *T. monococcum ssp. monococcum* (accession DV92) was investigated. Drought and heat effect examinations included seven treatments,as follows: control, drought 1 h, drought 6 h, heat 1 h, heat 6 h, and drought plus heat 1 h, drought plus heat 6 h. Powdery mildew pathogen stress: included non-innoculation, powdery 24 h, powdery 48 h and powdery 72 h.

### RNA extraction and real-time quantitative PCR

The total RNA was extracted from samples of fungal inoculated leaves using the optimized extraction procedure described by *Zhang et al. (2014)*.

The SYBR Green Premix Ex Taq^TM II quantitative PCR system (Takara, Dalian) was used for qPCR analysis. All experiments involving q-PCR were performed on a Q7 Real-Time PCR System (Applied Biosystems, Foster City, CA, USA). The actin gene (GenBank: aK458277.1) was used as the reference gene. The PCR reaction and program were modified according to the manual. The PCR reaction (a total reaction volume of 10 µL) comprised 5 µL 2× SYBR Green PCR Master Mix, 3 µL of the cDNA product, 1 µL of primer mix, and 1 µL of DNase/RNase-free water. The quantitative PCR thermal cycler program included 95 °C for 10 s, followed by 40 cycles at 95 °C for 5 s and 60 °C for 31 s. All primers for q-PCR were synthesized by the same company (AoKe, Yangling) (Table S4).

## RESULTS

### Identification of the SNARE protein in wheat

To identify SNARE proteins in wheat, the HMMER profile was implemented to identify the wheat genomes. The results showed that 173 hypothetical *TaSNARE* genes were characterized from wheat databases (Table S1). Qa, Qb, Qc, Qb+Qc and R SNARE proteins comprised 48(27.7%), 37(21.4%), 39(22.5%), 13(7.5%) and 36(20.8%) respectively. Among all 21 subfamilies, SYP1 contained a maximum of 33 proteins, and VAMP72 had the second most, at 15 proteins (Table S1). The encoded proteins comprised between 121

and 466 amino acid residues, the PIs ranged from 4.72 to 9.65, and the molecular weights were distributed from 13,687.37 to 51665.97 Da (Table S1).

All the sequences were divided into 64 groups in wheat (Table S1). Among these groups, 38 groups representing 114 genes contained three genes from each of the different subgenomes that were regarded as orthologous copies of a single *SNARE* gene named triplet. Five groups contained different homoeologous genes that were from the same homoeologous group (e.g., *TaSYP43-4AL*, *TaSYP43-7AS*, and *TaSYP43-7DS*). Eight groups contained two genes (e.g., *TaSYP131-2BS* and *TaSYP131-2DS*), and the remaining 8 groups consisted of only one gene (e.g., *TaGOS12-6BS*). Five groups had four genes, among which four groups had tandemly repeated genes (e.g., *TaSNAP1-2A1*, *TaSNAP1-2A2*, *TaSNAP1-2B*, and *TaSNAP1-2D*). We found that the Go term ''vesicle-mediated transport'' was most significantly enriched in the SNARE proteins (Table S2).

## Chromosomal locations and gene collinearity analysis of *SNARE* gene family members in wheat

Most collinear gene pairs occur within the same chromosome group (Fig. 1). The chromosomal distribution of the *SNARE* gene family of *T. aestivum* was analyzed. Figure 2 revealed the chromosomal location of 173 *SNARE* genes. All 21 wheat chromosomes have several *SNARE* gene family members: the wheat 1 to 7 homoeologous groups had 14 (1A = 5,1B = 4,1D = 5), 22 (2A = 7, 2B = 8, 2D = 7), 31 (3A = 12, 3B = 9, 3D = 10), 25 (4A = 9, 4B = 9 4D = 7), 23 (5A = 8, 5B=9, 5D=6), 26 (6A = 9, 6B = 9, 6D = 8) and 27 (7A = 11,7B = 8,7D = 9) SNARE genes, and 4 had no chromosomal location. In addition to homoeologous group 1, the *SNARE* genes were evenly distributed in the wheat genome, and the number of genes on each chromosome is similar. The most striking result to emerge from Fig. 2 was that the triplets, which were from different subgenomes, were similar in terms of their relative position on chromosomes.

## Phylogenetic, motif and structural analysis of the *SNARE* family genes

To further analyze the phylogeny, motif and structure of *TaSNARE*s, we selected one protein (genome group A chromosomal priority selection) from each of the 64 groups and obtained 64 SNAREs. The results showed that these proteins were primarily divided into 5 clades (Fig. 3). Most clades had three homoeologous proteins in the same branch, and these three homoeologous proteins were from three chromosomes in the same homoeologous group.

It is apparent from Fig. 3 that *SNARE*s in different subfamilies had different motifs. Qa had motifs 1, 2, 5, 8, 11 and 13. Qb had motifs 6, 7, 10, 12 and 13. Qc had motifs 5, 9 and 13. Qb+Qc had motifs 9 and 12. R had motifs 3, 4, 13 and 14. The results showed that motifs 6, 8, 10, 11, 13 and 15 were not predicted as being present in these SNAREs. Motif 1 was the SNARE domain; motifs 2 and 5 were syntaxin domains; motif 3 was the synaptobrevin domain, motifs 4 and 14 were longin domains; motif 7 was the SEC20 domain; and motif 12 was the V-SNARE-C domain. Qa, Qb, Qc, and R had motif 13, which was located in the C-terminus and associated SNAREs with lipid bilayers, and this motif was named the transmembrane (TM) domain (*Lipka, Kwon & Panstruga, 2007*).

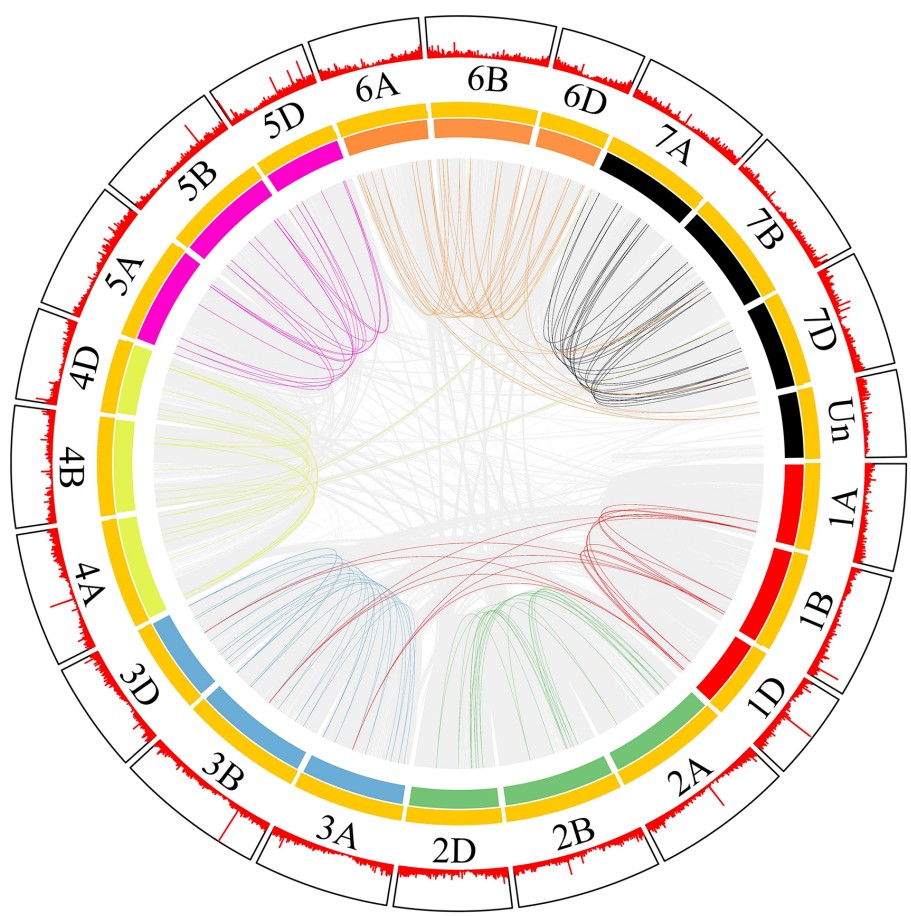

**Figure 1** *SNARE* **collinear gene pairs in wheat.** The innermost line represents a collinear gene pair, in which the *SNARE*s gene pair are coloured and the others are gray. The length of the innermost colored arc is the size of the chromosome. Each yellow line in the middle is a gene. The outer red line shows the gene density.

## Cis-acting elements of *TaSNARE* genes

Further analysis of the 2,000-bp promoter upstream of the 5′ end of the *TaSNARE* gene was performed. This promoter contains 9 types of resistance-related cis-acting elements (Table S3), including W-box (Cis-I), germ-related (Cis-II), MYB (Cis-III), SA-responsive (Cis-IV), Eth-responsive (Cis-V), EIRE (Cis-VI), G-box (Cis-VII), H-box (Cis-VIII) and IAA-responsive (Cis-IX) elements.

As shown in Fig. 4, Cis-I to Cis-IX were represented by 2230, 5054, 1647, 882, 170, 225, 152, 118 and 309 elements, respectively, in all 173 *SNARE* gene promoters. Cis-I, Cis-II and Cis-III made up 82.92% of all disease-related elements. Among the Cis-I elements, *TaUSE12-7A* was the most abundant (36). Among the Cis-II elements, *TaSEC222-5B* was the most abundant (87). Among the Cis-III elements, *TaSEC222-5A* was the most abundant (19).

In one triplet, for the promoters of the resistance-related elements, the numbers were similar. However, there were exceptions, as follows: *TaSFT11-2A/B/D* had 5/21/16 Cis-I

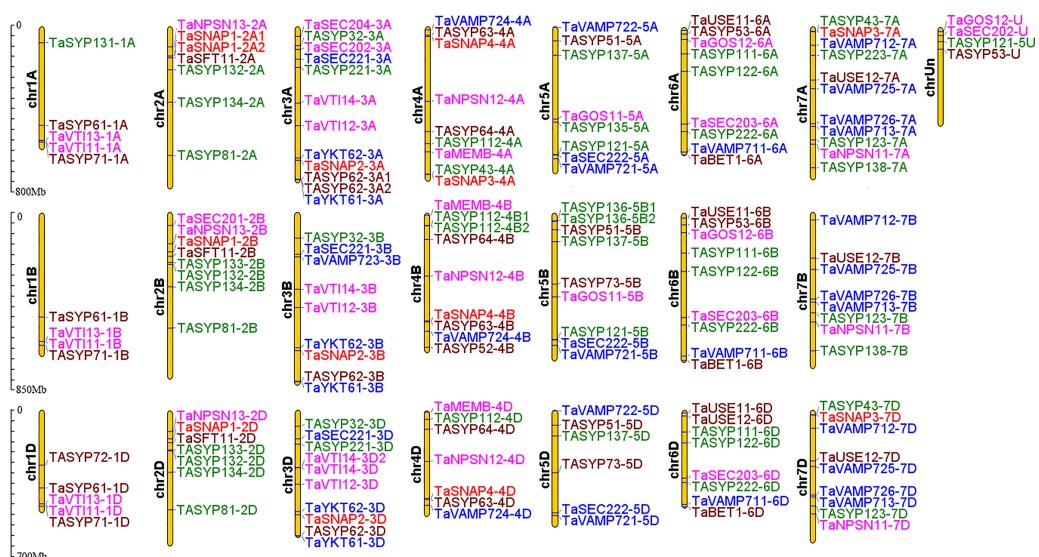

**Figure 2** **Chromosomal locations of *SNARE* genes in wheat.** A total of 173 *SNARE* genes were localized to *Triticum aestivum*. Qa SNARE: Green. Qb SNARE: Fuchsia. Qc SNARE: Brown. Qb+Qc SNARE: Red. *R SNARE*: Blue.

elements; *TaNPSN12-4A/B/D* had 71/17/21 Cis-II elements, and *TaSYP222-6A/B/D* had 3/11/1 Cis-IV elements.

## Expression analysis of *TaSNARE* genes by RNA-seq

To further understand the functions of the *SNARE* genes, we extracted gene expression information for 54 genes from six published RNA-seq databases (Fig. 5).

As shown in Fig. 5, in the growth period of wheat, the *SNARE* gene is expressed in roots, stems, leaves, seeds and spikes, showing low levels in seeds and leaves and high levels in roots, stems, and spikes. In the seeds of Z75, the expression levels of most genes (45) were very low, and in Z71-Z75-Z85, a high-low-high expression pattern was observed. Many genes (36) were most highly expressed in the 20 DPA aleurone layer during seed development. Most *SNARE* genes (49) are better expressed under light conditions than under dark conditions. Among these genes, *SYP122-6A* showed higher expression under light than in the dark in DV92, but in G3116, the opposite trend was observed. Compared with the control, the expression of 22 genes was upregulated 6 h after stress (drought 6 h, heat 6 h or drought plus heat 6h), and the expression patterns of 9 genes showed the opposite trend. In the process of leaf senescence, 42 genes showed the highest expression at 12 DAA. More than half of the genes (33) had the following expression distribution pattern in the grain layers: outer pericarp >inner pericarp >endosperm. In the powdery mildew pathogen stress, 3 genes were down-regulated by more than 0.5 times in 24 h, and one gene was up-regulated by more than 1 times. In total, 17 genes were up-regulated by more than 0.5 times, of which 8 genes were up-regulated by more than double. There were 6 genes down-regulated by more than 0.5 times at 48 h, and one gene was down-regulated by more than 1 times. There were 12 genes up-regulated by more than 0.5 times, of which

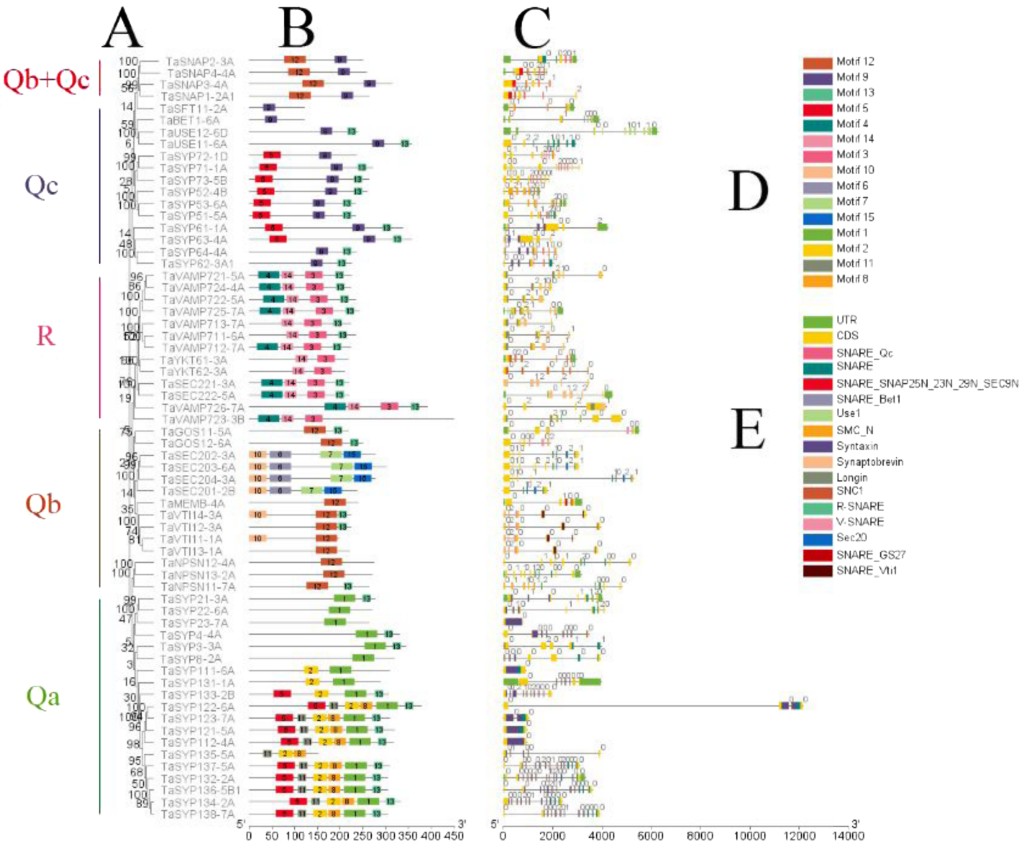

**Figure 3 Phylogenetic analysis, gene structure, domain location and motif compositions of the *SNARE gene* family in wheat.** (A) The clustering of genes in each subfamily. The 15 different color blocks in B represent different domains predicted in MEME, and (D) was shown in detail. (C) The distribution of gene exons and introns, and the position of the predicted domains in pfam in the exons; (E) domain name. The black lines in B and C indicate the length of the amino acid/gene sequence.

five genes were up-regulated by more than 1 time. At 72 h, 5 genes were down-regulated by 0.5 times, of which 2 genes were down-regulated by more than 1 time. There were 13 genes up-regulated by more than 0.5 times. There were 6 genes that were continuously up-regulated between 24 h–72 h. *TaSYP135* continued to be down-regulated between 24 h–72 h. These genes with the most dramatic changes in expression may have played a role in responding to powdery mildew infection.

## Expression patterns of *TaSNARE* genes under powdery mildew treatment

We selected one gene from each of the 21 classes, and we obtained 21 *TaSNARE* genes (*TaYKT6* had no signal) specific to the designed primers (Table S3). As shown in Fig. 6, the expression patterns of different *SNARE* genes in the same sample and subfamily were similar. Most of the *TaSNARE* genes had similar expression pattern in 7M US and CS, but in 7M CH, a different expression pattern was observed. A majority of the *TaSNARE* genes

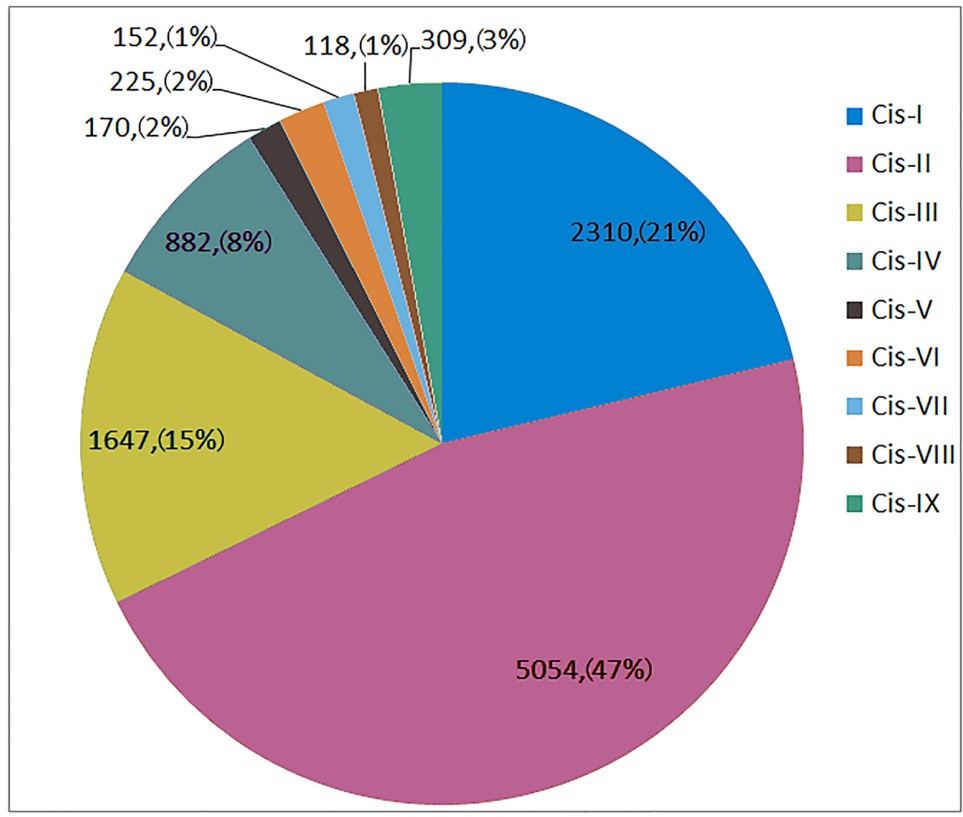

**Figure 4** **The number and proportion of 9 disease-related cis-acting element in *SNARE* genes promoter.** Cis-I: W-box; Cis-II:Germs-related; Cis-III:MYB; Cis-IV:SA responsible; Cis-V:Eth responsible; Cis-VI:EIRE; Cis-VII:G-box; Cis-VIII:H-box; Cis-IX:IAA responsible.

in 7M CH had high expression levels at 6 h. *TaSYP4*, *TaSYP8*, *TaMEMB* and *TaSEC22* had high expression levels at 6 h in 'Shaanyou 225' but not in the other wheat.

In the Qa subfamily, the expression of all genes changed little at each time point in CS. *SYP121*, *SYP221*, and *SYP3* were upregulated in the 7M CH 6 h sample but downregulated in 'Shaanyou 225'. *SYP4* and *SYP8* were upregulated in the 'Shaanyou 225' 6 h sample but not in the 7M CH sample. *QaSNARE* expression was similar between 7M US and 'Shaanyou 225'. This may indicate that the up-regulated expression of *SYP4* and *TaSYP8* played a negative role in the wheat response to powdery mildew infection.

In the Qb subfamily, *GOS12* expression was upregulated at 6 h and then downregulated in four wheat varieties. *MEMB* expression was upregulated in 'Shaanyou 225' at 6 h, 24 h, and 48 h but not in CS. There was no significant difference in the expression at different time points in 7M CH and 7M US. The *VTI12* expression patterns were similar to those of MEMB in 'Shaanyou 225' and 7M CH. *NPSN11* expression in four wheat varieties were similar, showing downregulation at 6 h–48 h, except for the upregulation at 24 h–48 h observed in 7M CH. *SEC203* was downregulated at 6h-48 h in 7M US and CS. This gene was downregulated at 12 h–24 h and upregulated at 48 h in 'Shaanyou 225'. In 7M CH, *SEC203* was downregulated at 24 h, and the other genes were upregulated. This shows that

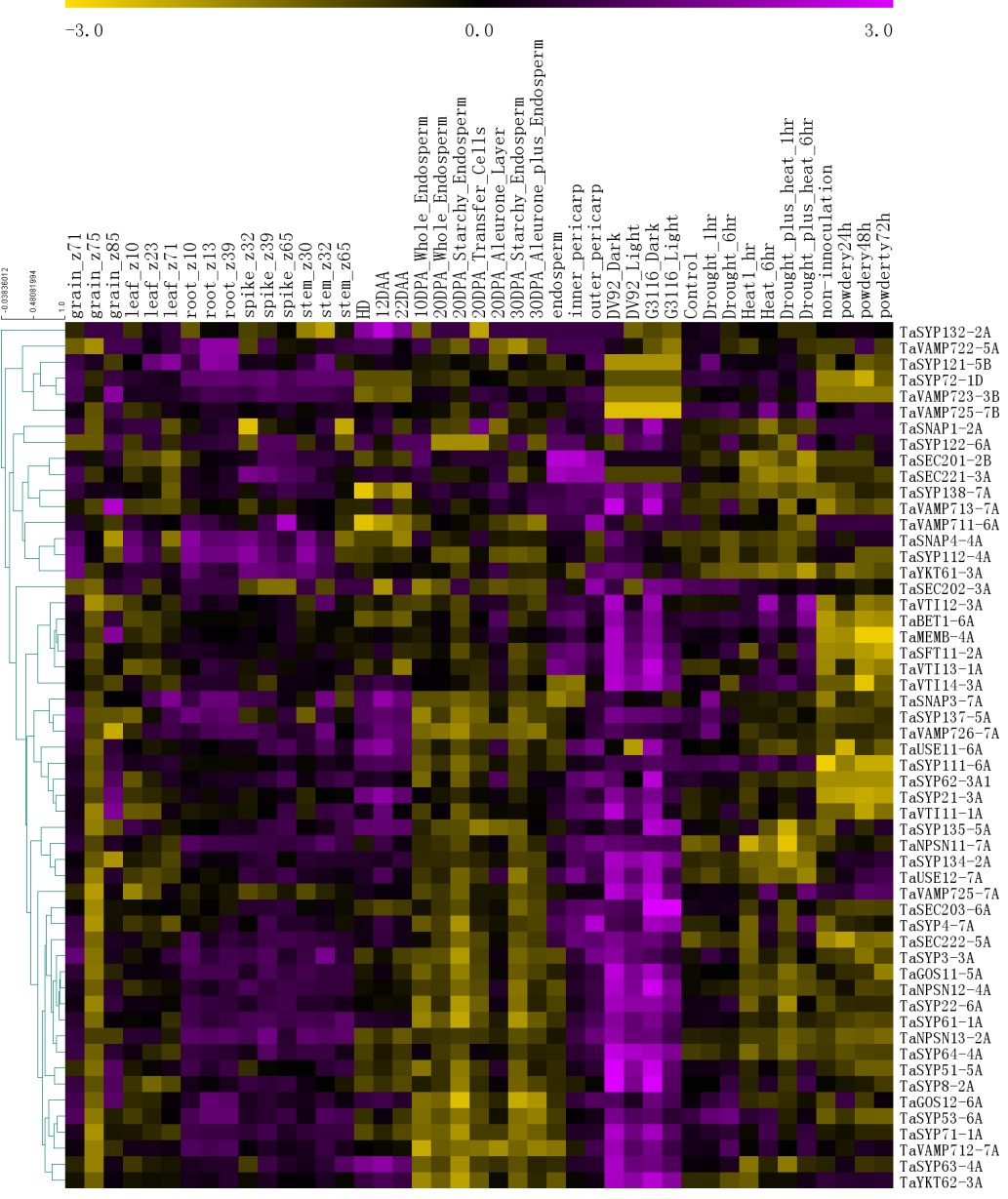

**Figure 5** **The expression profiles of *TaSNARE* genes in deferent treatment and stage.** Developmental time course: Z10-Z85. Grain layers at 12 DPA: outer pericarp, inner pericarp and endosperm. Grain layer developmental time course: 10 DPA whole endosperm, 20 DPA whole endosperm, 20 DPA starchy endosperm, 20 DPA transfer cells, 20 DPA aleurone, 30 DPA starchy endosperm, 30 DPA aleurone plus endosperm. Senescing leaf time course: HD, 12 DAA and 22 DAA. Photomorphogenesis for DV92 and G3116. Drought and heat: control, drought 1 h, drought 6 h, heat 1 h, heat 6 h, drought plus heat 1 h, drought plus heat 6 h. Powdery mildew pathogen stress: included non-innoculation, powdery24 h, powdery 48 h and powdery 72 h.

the *NPSN11* gene has little effect in the early stage of wheat's response to powdery mildew infection, while *GOS12* and *SEC203* play a certain role.
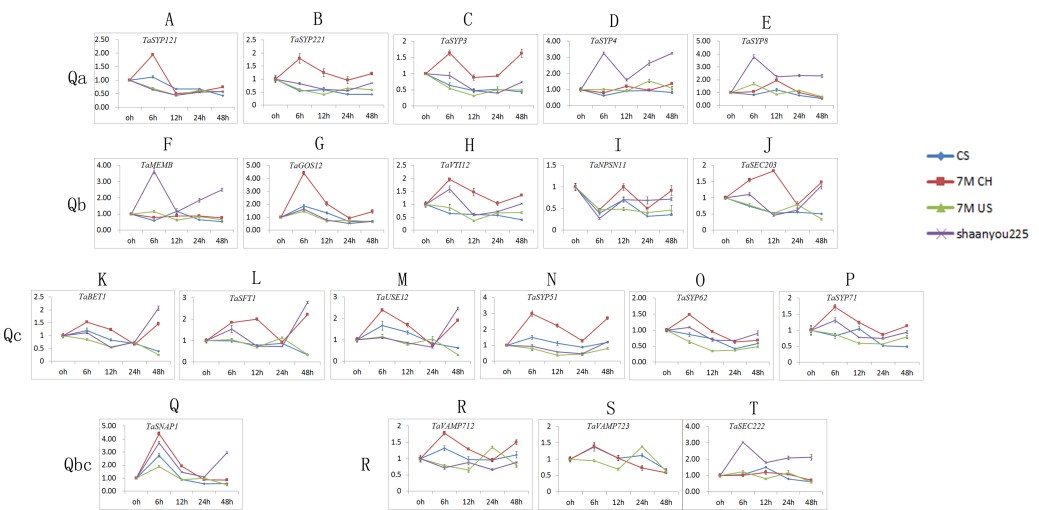

**Figure 6** *TaSNARE* **genes expression patterns infected by Bgt E09 (A–T).** CS: susceptible, Chinese spring. 7M CH: resistant, NA0973-5-4-1-2-9-1 (CS-SY159 DA 7Mg, (CS)/ *Ae. geniculata* SY159//CS)). 7M US: susceptible, TA7661 (CS-AEGEN DA 7Mg). Shaanyou225: susceptible, wheat cultivate variety Shaanyou225.

In the Qc subfamily, all *QcSNARE*s in the same variety were similar. In CS and 7M US, most genes were downregulated at 6–24 h. In 7M CH, certain genes were downregulated at 24 h, while the others were upregulated. In 'Shaanyou 225', the genes were downregulated at 12 h–24 h and upregulated at 6 h and 48 h. This means that after powdery mildew infects wheat for 48 h, *BET1*, *SFT1*, *USE12*, and *SYP5* were resistant to 7M CH up-regulation and susceptible to 7M US down-regulation, which implies that the up-regulation of these genes has a positive effect in these two types of wheat.

In the Qb+Qc subfamily, *SNAP1* was upregulated at 6 h in all varieties and downregulated at 12 h–48 h, except in 'Shaanyou 225' at 48 h, in which *SNAP1* was upregulated. This may indicate that the *SNAP1* gene may play a similar role in the four types of wheat.

In the R subfamily, in CS, *VAMP712* was upregulated at 6 h and at the other time points, it showed no change. In 'Shaanyou 225', *VAMP712* was downregulated at 6 h and 24 h, and at the other time points there was no change. In 7M CH, *VAMP712* was upregulated at 6 h, 12 h, and 48 h and showed no change at 24 h. In 7M US, *VAMP712* was upregulated at 24 h and downregulated at the other time points. No signal for *VAMP723* was detected in 'Shaanyou 225'. In 7M US, *VAMP723* was upregulated at 24 h and downregulated at other time points. In 7M CH, *VAMP723* was upregulated at 12 h and downregulated at 24 h and 48 h. In CS, *VAMP723* was upregulated at 6 h and downregulated at 48 h. *SEC222* was upregulated at every time point in 'Shaanyou 225', and the other varieties showed no significant difference. This implies that the up-regulated expression of *SEC222* plays a negative role in the powdery mildew infection of 'Shaanyou 225'.

With the evolution of plants, wheat *SNARE* genes are constantly changing. *SNARE* genes have different expression patterns after being infected by powdery mildew, suggesting that

these genes may be involved in the biological stress response to powdery mildew in different aspects.

## DISCUSSION

*SNARE*s are mainly involved in membrane-related life activities. It is apparent from previous reports that *SNARE*s are rarely described from the perspective of gene families.

The completion of the wheat genome sequencing work will help in the analysis of key genes and agronomic traits of wheat at the genomic level. However, the sequencing work has progressed slowly because of the very large size of the genome and the high number of repeated sequences. In this paper, 173 nonredundant *SNARE* genes were obtained from the newly published IWGSC 1.0 wheat genome reference sequence. In another study, using more stringent identification methods, 27 *SNARE* and 8 *NPSN* genes were discovered (*Gaggar, Kumar & Mukhopadhyay, 2020*). Common wheat is a heterogeneous hexaploid crop, and it usually contains three paralogous homoeologous genes from groups A, B, and D, which can be called a triplet. However, in our study, 16 out of 64 groups did not appear as triplets (Table S1). This could be explained by the loss of these genes during long-term evolution, or it could be due to insufficient sequencing depth or incomplete splicing. There are also some triplets in which *A/B/U* occurs, possibly because the difficulty in splicing leads to the genes not being appropriately located in their chromosomes.

As evidenced by the analysis of cis-acting elements, genes in the same triad are most alike in their components. However, there remain a few differences, which may lead to some bias in the expression of these homoeologous genes in some physiological states. In addition, depending on the composition of cis-acting elements found, the genes mainly contained W-boxes, disease-related elements and MYBs. This suggested that *SNARE* resistance in plants may mainly be regulated by transcription factors such as WRKY, MYB and other disease resistance genes.

Some interesting information was obtained by analyzing the RNA-seq data. Because these RNA-seq databases are relatively old, the genetic information used was a genetic sketch of wheat. We compared the TGAC v1.0 data to a sketch database to find the corresponding *SNARE* gene in the sketch. The expression patterns of triplet genes in the same group were very similar, and in photomorphogenesis, most of the triplets did not show gene expression data for the B and D genomes, so we selected the A genome in the triplet. The gene was analyzed, and if there was no group A gene, a gene of group B or D was used.

Members of the same class as the subfamilies have diverse roles in the same life activities. In *Arabidopsis*, severe male gametophytic defects occur only when *syp123*, *syp125*, and *syp131* were simultaneously mutated (*Slane et al., 2017*). *Arabidopsis* SCYL2B and CHC1 undergo vesicle transport through *VTI11* or *VTI12* for plant growth (*Jung et al., 2017*). On the other hand, homoeologous genes may also play different roles. Overexpression of *OsVAMP7111* did not enhance rice resistance to blast, while overexpression of *OsVAMP714* increased the resistance. This suggests that *VAMP714* is potentially specific for resistance to rice blast (*Sugano et al., 2016*). PEN1 in plants forms the SNARE complex with VAMP721

and *VAMP722* during defense against powdery mildew fungi, and it also forms SNARE complexes in vitro with VAMP724 and VAMP727, which are not related to plant immunity (*Kwon et al., 2008*). *PVA31* participates in SA-associated apoptosis by interacting with *VAMP721/722/724* but not *VAMP711/727* to combat pathogen infection (*Ichikawa et al., 2015*). In wheat, silencing of *TaNPSN11/13* reduced the resistance to *CYR23*, whereas silencing of *TaNPSN12* did not have the same effect (*Wang et al., 2014*).

The RNA-seq data showed that homoeologous genes in the same evolutionary branch exhibit many different expression patterns under the same conditions, as observed for *VTI11/12/13/14* and *GOS11/12*, but there were some differences. For example, *NPSN11/13* were in one class, and *NPSN12* was not clustered with *NPSN11/13*. In the senescing leaf time course, *NPSN11* and *NPSN13* exhibited low-high-low expression patterns, while *NPSN12* showed no difference in expression at each stage. During photomorphogenesis, the expression of *NPSN12* and *NPSN13* in the dark was higher than that in the light, and that of *NPSN11* showed no difference. In the heat and drought treatments, both *NPSN12/13* were downregulated compared to the control, and there was no change in expression from 1 to 6 h after treatment. On the other hand, after treatment, *NPSN11* was upregulated under drought, upregulated at high temperature and upregulated 1 h–6 h after treatment. In another group of subfamily genes, the *SNAP* genes, the expression patterns differed greatly among the three members. The expression level of *SNAP3* (FPKM) was higher than that of *SNAP1/4* in each period and process. In the developmental time course of wheat, *SNAP3* was the most highly upregulated in all tissues at various developmental stages, while *SNAP4* was downregulated, and *SNAP1* expression was low. In the grain layers, the expression of *SNAP1* had the distribution endosperm>outer pericarp>inner pericarp, *SNAP2* expression had the distribution endosperm=inner pericarp>outer pericarp, and *SNAP4* expression had the distribution outer pericarp>inner pericarp >endosperm. Under heat and drought and in the senescing leaf time course, only *SNAP3* expression was high, while the other FPKM values were less than 1. After powdery mildew infection, the RNA-seq results were different from the quantitative results. This could be explained from the following aspects. First, there was a big difference between the materials of this study and (*Zhang et al., 2014*). The N9134 material was the offspring of tetraploid durum wheat, and this study, CS additional lines were formed by crossing with *Ae. geniculata*. Second, in the time of expression change, most of the resistant materials that we quantified reached their peak at 6 h, while RNA-seq reached the peak at 24 h. The expression in 24 h was not clear. However, the expression level of this family gene will be changed by the signals from powdery mildew infection, suggesting that *SNARE* plays a certain role.

CS is the parent of A and B, and the other parents are different varieties of *Ae. geniculata*. However, their resistances to powdery mildew are quite different. We chose these two materials to try to explain the effects of exogenous chromosomes on endogenous gene expression from a genomic perspective. Figure 6 shows that the 7M US and CS expression patterns were similar after infection with powdery mildew, but the expression pattern of 7M CH was very different from those of the other two varieties. This suggests that our exogenous chromosomes had some effect on the endogenous gene expression and may have led to differences in resistance. It has been reported that after the introduction of exogenous

chromosomes, genes on exogenous 7M$^g$ chromosomes mainly affect homologous genes on homologous chromosomes. The resistance gene carried on 7M $^g$ may affect the expression of *SNARE*-related genes. Therefore, we propose two hypotheses. The first is that the resistance gene of exogenous 7M$^g$ could resist powdery mildew by participating in the disease resistance pathway of wheat. Second, it is possible that the exogenous 7M$^g$ chromosome achieves resistance to powdery mildew by affecting the expression of the endogenous seventh homologous gene.

In conclusion, this paper identified 173 *SNARE*s in wheat, which laid a foundation for further studies on the function of *SNARE* genes. In addition, these results will also be helpful for further study of the powdery mildew resistance of wheat.

### Funding

This work was supported by the National Key Research and Development Plan (2016YFD0102004). The funders had no role in study design, data collection and analysis, decision to publish, or preparation of the manuscript.

### Grant Disclosures

The following grant information was disclosed by the authors:
National Key Research and Development Plan:
2016YFD0102004.

### Competing Interests

The authors declare there are no competing interests.

### Author Contributions

- Guanghao Wang conceived and designed the experiments, performed the experiments, analyzed the data, prepared figures and/or tables, authored or reviewed drafts of the paper, and approved the final draft.
- Deyu Long conceived and designed the experiments, performed the experiments, analyzed the data, prepared figures and/or tables, and approved the final draft.
- Fagang Yu performed the experiments, prepared figures and/or tables, and approved the final draft.
- Hong Zhang analyzed the data, prepared figures and/or tables, and approved the final draft.
- Chunhuan Chen conceived and designed the experiments, authored or reviewed drafts of the paper, and approved the final draft.
- Yajuan Wang conceived and designed the experiments, prepared figures and/or tables, and approved the final draft.
- Wanquan Ji conceived and designed the experiments, prepared figures and/or tables, authored or reviewed drafts of the paper, and approved the final draft.

## Data Availability

The raw measurements are available in the Supplementary Files.

## Supplemental Information

Supplemental information for this article can be found online at http://dx.doi.org/10.7717/peerj.10788#supplemental-information.

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
