# Peer review of "Genome-wide identification, evolution, and expression of the SNARE gene family in wheat resistance to powdery mildew"

_PeerJ, doi:10.7717/peerj.10788_

## Round 0.1 · original submission · Minor Revisions

Sorry for the delay with the decision. Some reviewers could not reply in time. We have received 3 reviews now. Some remarks demand major revision. But I believe, the comments are not very critical. Please check the presentation of the figures. Possibly it needs to update the figures as suggested by reviewer #2. Waiting for resubmission of the revised manuscript.

·

Basic reporting

I had gone through the manuscript entitled 'Genome-wide identification, evolution, and expression of SNARE genes family in wheat'. The manuscript is well written using good English language. The authors also provided background information on SNAREs, still they can include the following two papers under discussion and mention a few lines:
1.Gaggar P, Kumar M, Mukhopadhyay K (2020) Genome-scale identification, in silico characterization and interaction study between wheat SNARE and NPSN gene families involved in vesicular transport. IEEE/ACM Transactions on Computational Biology and Bioinformatics. DOI: 10.1109/TCBB.2020.2981896
2. Wang X, Wang X, Deng L, Chang H, Dubcovsky J, Feng H, Han Q, Huang L, Kang Z, (2014). Wheat TaNPSN SNARE homologues are involved in vesicle-mediated resistance to stripe rust (Puccinia striiformis f. sp. tritici). Journal of experimental botany, 65,4807-4820.

Experimental design

All the experiments are properly designed. Aim of the manuscript clearly mentioned

Validity of the findings

A bit lenient method had been followed for identification of the genes from wheat genome, which is acceptable. A more stringent method would have been better, but that will reduce the number of identified genes.

Additional comments

The following aspects should be rectified:
1. When all authors belong to the same lab of the same University, there is no need to add superscript 1 after all names
2. Line 43 and 361 the author names should be: Snyder BA and Nicholson RL
3. Line 64 and 73: Italicize the pathogen names Phytophthora infestans and Magnaporthe oryzae
4. Line 108: Remove one set of the two sets of parenthesis.
5. Lines 184-185: Explain why the colinear gene pairs do specifically exist on adjascent chromosomes. Has similar situations found with any other genes in wheat?
6. Lines 220-223: What are A,B and C elements, please explain. In Fig. 4 also, as referred in the text, nothing is mentioned about these elements.
7. Line 301: Capitalize t after full stop
8 Many article titles in the reference list has been copied from original article showing capitalized first letters. Please provide running titles.

·

Basic reporting

There are numerous grammatical mistakes, misspell words which needs to be checked thoroughly by a native English speaker, otherwise not considered for publication in PeerJ.

Line 38-39 scientific names not written in italic.

Experimental design

The title of the manuscript should include expression analysis of snare genes in wheat against powdery mildew infection to make it more sound.

The SNARE protein full form is not mentioned once in the manuscript.

In introduction part no background is given on the fungus used in the study as well as why was this fungus chosen for experimental work.

Line 88 says all plant snare proteins sequences, which plants, name them or add a supplementary file.

In section 2.5 kindly mention clearly the time points of sample collection after fungus treatment.

Which method was used for powdery mildew conidia infection on wheat is missing.

Validity of the findings

Fig. 6 font needs to be increased. As each row in the fig depicts one family of SNARE eg first row is of Qa snare then this fig can be made more clear by separating each family so that patterns are visible among each family.

In Section 3.6 the expression results are reported in a straight manner and the inference of the expression data obtained is missing.

Why did the authors choose to do a RNA seq expression of SNARE genes which reveal only abiotic stress like drought and heat and no biotic stress like fungal infection time points as only the later was focused in this study. The RNA seq expression profile and the Qrt-pcr results does not tally to give a conclusion on snare roles in biotic stress. Please explain.

In Fig 4, the pie charts shows 882,8%, 1647,15% and so on, this is confusing and should be written as 882 (8%) and so on.

·

Basic reporting

I carefully read the text of the manuscript «Genome-wide identification, evolution, and expression of SNARE genes family in wheat.» Overall, it describes a perfect study, but the authors should correct the style and mistakes/typos in English.
Also, I think you should expand your keyword list by adding words that more closely specify the research.
Line 77-81: The last paragraph of the introduction can be expanded to indicate the article's purpose and main results more accurately.
The full form for the abbreviation SNARE must be indicated at the beginning of the introduction.

Experimental design

In the introduction part, the authors should introduce more information about the fungus used in the study and specify why they used this fungus for experimental work.
The authors should check the methodology described in "Fungus and Wheat materials": the method of infection and how long after infection did they fix the research material?

Validity of the findings

no comment

Additional comments

Line 64 and 73: The pathogen names ''Phytophthora infestans'' and ''Magnaporthe oryzae'' should be italicized.
Fig 4. Please, check the % values, you should divide all values by 100.
Fig6. It is necessary to increase the font of the axis labels.

---

## Round 0.2 · Minor Revisions

There are some remaining minor remarks from reviewer #1 and #2. Please answer point-by-point and resubmit the manuscript soon.

·

Basic reporting

English language still needs polishing, still there are a few spelling mistakes and grammatical errors. I'm mentioning the following few:
Line 15: Phylogenetic tree
Line 99: Synaptobrevin, not synaptebrevin
Line 135:Use growth chamber instead of incubator
Line 137: method
Line 138: use collected instead of cut
Line 174: Actin gene; also provide the NCBI accession number within parenthesis
Line 211: in terms of their relative position on chromosomes
Line 374: SNAP4, P is missing
Line 382: Provide the publication year after Zhang et al., as there are two references with Zhang
Line 384: use we instead of I

All gene names should be italicized

There is a huge confusion between homologous and homoeologous at various places in the manuscript.

Experimental design

OK

Validity of the findings

Very good

Additional comments

English language still needs polishing, still there are spelling mistakes and grammatical errors.

·

Basic reporting

In the manuscript wherever this kind- Sun et al (Sun et al. 2017) of reference is put can be simply replaced by Sun et al. 2017.

In section 2.5 writing should be uniform for eg. at some points 24h is written whereas others are written as 48 h. Correct all as including space or no space in between. Apply this to all.

Line 153 omit extra bracket.

Line 186: Qa, Qb, Qc, Qb+Qc and R SNARE proteins have 48(27.7%), 187 37(21.4%), 39(22.5%), 13(7.5%) and 36(20.8%) respectively can be replaced by Qa, Qb, Qc, Qb+Qc and R SNARE proteins comprised 48(27.7%), 187 37(21.4%), 39(22.5%), 13(7.5%) and 36(20.8%) respectively.

Line 204:The results revealed 173 SNARE genes in the chromosomal location information (Fig 2) can be changed to Fig. 2 reveals the chromosomal location of 173 SNARE genes.

Experimental design

Section 2.2 mentions that maximum number of 10 motifs were taken whereas in Fig 3 15 number of motifs are obtained. 10 should be replaced with 15.

Validity of the findings

No comment

·

Basic reporting

The manuscript has been revised according to the comments of the reviewers and the text and figures have been greatly improved. In addition, the authors additionally corrected grammar.

Experimental design

no comment

Validity of the findings

no comment

Additional comments

I think you have done a good job and I think the editor should accept it for publication.

---

## Round 0.3 · Minor Revisions

Thanks for the manuscript update. The reviewers have no more critical remarks.

However, Gerard Lazo, the Section Editor, has commented and said:

As the authors strive to describe the SNAREs from the perspective of gene families, and to use more stringent identification, the manuscript fails to go the additional step in providing appropriate annotation to go with their classification. There were 173 SNAREs identified and sorted by tissue expression, developmental stage, and stress treatments. Gene ontologies are available to create a resource for pinning down gene function; this was not used.

Journal manuscripts are often scanned by text-mining software that locates and extracts core data elements, like gene function. Adding standard ontology terms, such as the Gene Ontology (GO, geneontology.org) or others from the OBO foundry (obofoundry.org) can enhance the recognition of your contribution and description. This will also make human curation of literature easier and more accurate. None of this was visible.

There were no tables provided which provided guidance to the sequences annotated; the coordinates should be included with table S1. Or possibly somewhere in the text which would leaded to a readily downloadable version of the 173 sequences, or just provide a FASTA file. It’s best to make the experience easier for the reader.

Please consider the suggestions as I would suggest moderate revision to clarify the dataset. RAR files are not that common; perhaps another format may be preferable; like tar and gzip to make *.tar.gz files. Below are some suggested revisions detected within the text:

EDITS
LINE NO: / BEFORE / AFTER / [COMMENTS]
LINE 15: / of SYP1 class was / of the SYP1 class type was / [.]
LINE 15: / Phylogenetic tree / Phylogenetic tree analysis / [.]
LINE 87: / without headed / without heading / [.]
LINE 89: / Phylogenetic tree / Phylogenetic tree analysis / [.]
LINE 140: / put them into / put into / [.]
LINE 140: / leaves are / leaves were / [.]
LINE 175: / acting gene / actin gene / [.]
LINE 215: / (A chromosomal / (genome group A chromosomal / [?]

Regarding Gene Ontology - you may take an alternative system. But all the positions should be documented.

Please revise and resubmit the manuscript soon, by this year.

---

## Round 0.4 · accepted · Accept

Thanks for the update. I have no more remarks.